# GENERALIZATION THROUGH MEMORIZATION: NEAREST NEIGHBOR LANGUAGE MODELS

**Urvashi Khandelwal**[†,∗] **Omer Levy**[‡]**, Dan Jurafsky**[†]**, Luke Zettlemoyer**[‡] **& Mike Lewis**[‡]
[†]Stanford University
[‡]Facebook AI Research
{urvashik,jurafsky}@stanford.edu
{omerlevy,lsz,mikelewis}@fb.com

## ABSTRACT

We introduce $k$NN-LMs, which extend a pre-trained neural language model (LM) by linearly interpolating it with a $k$-nearest neighbors ($k$NN) model. The nearest neighbors are computed according to distance in the pre-trained LM embedding space, and can be drawn from any text collection, including the original LM training data. Applying this augmentation to a strong WIKITEXT-103 LM, with neighbors drawn from the original training set, our $k$NN-LM achieves a new state-of-the-art perplexity of 15.79 – a 2.9 point improvement with no additional training. We also show that this approach has implications for efficiently scaling up to larger training sets and allows for effective domain adaptation, by simply varying the nearest neighbor datastore, again without further training. Qualitatively, the model is particularly helpful in predicting rare patterns, such as factual knowledge. Together, these results strongly suggest that learning similarity between sequences of text is easier than predicting the next word, and that nearest neighbor search is an effective approach for language modeling in the long tail.

## 1 INTRODUCTION

Neural language models (LMs) typically solve two subproblems: (1) mapping sentence prefixes to fixed-sized representations, and (2) using these representations to predict the next word in the text (Bengio et al., 2003; Mikolov et al., 2010). We present a new language modeling approach that is based on the hypothesis that the representation learning problem may be easier than the prediction problem. For example, any English speaker knows that *Dickens is the author of* and *Dickens wrote* will have essentially the same distribution over the next word, even if they do not know what that distribution is. We provide strong evidence that existing language models, similarly, are much better at the first problem, by using their prefix embeddings in a simple nearest neighbor scheme that significantly improves overall performance.

We introduce $k$NN-LM, an approach that extends a pre-trained LM by linearly interpolating its next word distribution with a $k$-nearest neighbors ($k$NN) model. The nearest neighbors are computed according to distance in the pre-trained embedding space and can be drawn from any text collection, including the original LM training data. This approach allows rare patterns to be memorized explicitly, rather than implicitly in model parameters. It also improves performance when the same training data is used for learning the prefix representations and the $k$NN model, strongly suggesting that the prediction problem is more challenging than previously appreciated.

To better measure these effects, we conduct an extensive empirical evaluation. Applying our $k$NN augmentation to a strong WIKITEXT-103 LM using only the original dataset achieves a new state-of-the-art perplexity of 15.79 – a 2.86 point improvement over the base model (Baevski & Auli, 2019) – with no additional training. We also show that the approach has implications for efficiently scaling up to larger training sets and allows for effective domain adaptation, by simply varying the nearest neighbor datastore. Training a model on 100-million tokens and using $k$NN search over a 3-billion token dataset can outperform training the same model on all 3-billion tokens, opening a

---

∗Work done while the first author was interning at Facebook AI Research.

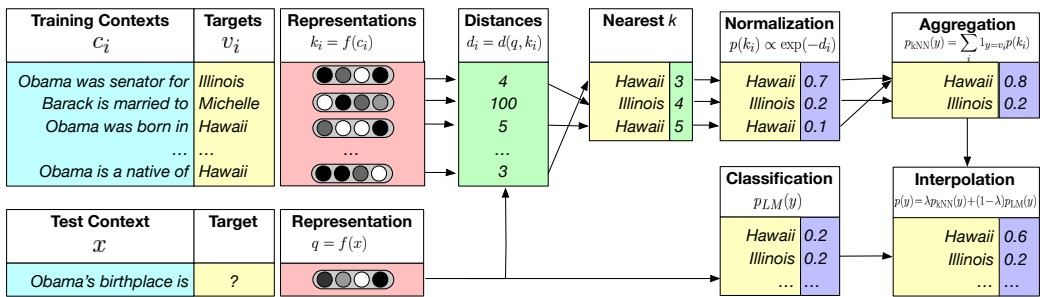

Figure 1: An illustration of $k$NN-LM. A datastore is constructed with an entry for each training set token, and an encoding of its leftward context. For inference, a test context is encoded, and the $k$ most similar training contexts are retrieved from the datastore, along with the corresponding targets. A distribution over targets is computed based on the distance of the corresponding context from the test context. This distribution is then interpolated with the original model's output distribution.

new path for efficiently using large datasets in language models. Similarly, adding out-of-domain data to the datastore makes a single LM useful across multiple domains, again without further training. Qualitatively, we find the model is particularly helpful for long-tail patterns, such as factual knowledge, which might be easier to access via explicit memory.

## 2 NEAREST NEIGHBOR LANGUAGE MODELING

Language models (LMs) assign probabilities to sequences. Given a *context* sequence of tokens $c_t = (w_1, \dots w_{t-1})$, autoregressive LMs estimate $p(w_t|c_t)$, the distribution over the *target* token $w_t$.

The $k$NN-LM involves augmenting such a pre-trained LM with a nearest neighbors retrieval mechanism, without any additional training (the representations learned by the LM remain unchanged). This can be done with a single forward pass over a text collection (potentially including the original LM training set), where the resulting context-target pairs are stored in a key-value datastore that is queried during inference, as illustrated in Figure 1.

**Datastore**    Let $f(\cdot)$ be the function that maps a context $c$ to a fixed-length vector representation computed by the pre-trained LM. For instance, in a Transformer LM, $f(c)$ could map $c$ to an intermediate representation that is output by an arbitrary self-attention layer. Then, given the $i$-th training example $(c_i, w_i) \in \mathcal{D}$, we define the key-value pair $(k_i, v_i)$, where the key $k_i$ is the vector representation of the context $f(c_i)$ and the value $v_i$ is the target word $w_i$. The datastore $(\mathcal{K}, \mathcal{V})$ is thus the set of all key-value pairs constructed from all the training examples in $\mathcal{D}$:

$$(\mathcal{K}, \mathcal{V}) = \{(f(c_i), w_i)|(c_i, w_i) \in \mathcal{D}\} \tag{1}$$

**Inference**    At test time, given the input context $x$ the model generates the output distribution over next words $p_{\text{LM}}(y|x)$ and the context representation $f(x)$. The model queries the datastore with $f(x)$ to retrieve its $k$-nearest neighbors $\mathcal{N}$ according to a distance function $d(\cdot, \cdot)$ (squared $L^2$ distance in our experiments, making the similarity function an RBF kernel).Then, it computes a distribution over neighbors based on a softmax of their negative distances, while aggregating probability mass for each vocabulary item across all its occurrences in the retrieved targets (items that do not appear in the retrieved targets have zero probability):

$$p_{\text{kNN}}(y|x) \propto \sum_{(k_i, v_i) \in \mathcal{N}} \mathbb{1}_{y=v_i} \exp(-d(k_i, f(x))) \tag{2}$$

Finally, we follow Grave et al. (2017a) and interpolate the nearest neighbor distribution $p_{\text{kNN}}$ with the model distribution $p_{\text{LM}}$ using a tuned parameter $\lambda$ to produce the final $k$NN-LM distribution:

$$p(y|x) = \lambda\, p_{\text{kNN}}(y|x) + (1 - \lambda)\, p_{\text{LM}}(y|x) \tag{3}$$

**Implementation**    The datastore contains an entry for each target in the training set, which for LMs can be up to billions of examples. To search over this large datastore, we use FAISS (Johnson et al., 2017), an open source library for fast nearest neighbor retrieval in high dimensional spaces. FAISS speeds up search by clustering the keys and looking up neighbors based on the cluster centroids, while reducing memory usage by storing compressed versions of the vectors. We found in preliminary experiments that using $L^2$ distance for FAISS retrieval results in better performance for $k$NN-LM, compared to inner product distance.

**Related Cache Models**    Prior work (Grave et al., 2017c; Merity et al., 2017) used a similar approach to compute similarity to the previous hidden states of *test* documents, making it easier to copy rare vocabulary items from the recent past. Such techniques have been less popular since the development of Transformers (Vaswani et al., 2017), which can learn to copy recent words using self-attention; in Section 4.1, we observe relatively small gains from caching recent items in the same test document à la Grave et al. (2017c). Most relatedly, Grave et al. (2017a) describe an *online* language model using nearest neighbor search over all previous hidden states, to improve domain adaptation. In our work, we only save training data, with the goal of explicitly memorizing training examples to better generalize to similar cases at test time.

# 3    EXPERIMENTAL SETUP

**Data**    Experiments in this paper use the following English corpora:

WIKITEXT-103 is a standard benchmark by Merity et al. (2017) for autoregressive language modeling with a 250K word-level vocabulary. It consists of 103M tokens of Wikipedia in the training set and 250K tokens in each of the development and test sets.

BOOKS is the Toronto Books Corpus (Zhu et al., 2015), containing 0.7B. Complete books are held out for validation/test.

WIKI-3B is English Wikipedia, containing about 2.87B tokens. Whole articles are held out for validation/test.

WIKI-100M is a random 100M token subset of WIKI-3B, consisting of complete articles.

Except for WIKITEXT-103, text is tokenized using the byte-pair encoding (Sennrich et al., 2015) with the 29K subword vocabulary from BERT (Devlin et al., 2019).

**Model Architecture**    $k$NN-LM is compatible with any model that produces fixed size context representations. We use decoder-only Transformers (Vaswani et al., 2017) for language modeling, which are the current state of the art. Since the $k$NN-LM makes no changes to the underlying LM, we take the exact architecture and optimization described by Baevski & Auli (2019) and use it to create a $k$NN-LM for inference. This model consists of 16 layers, each with 16 self-attention heads, 1024 dimensional hidden states, and 4096 dimensional feedforward layers, amounting to 247M trainable parameters. It processes 3072 tokens of context per example for WIKITEXT-103 and 1024 tokens for the rest of the corpora. Following Baevski & Auli (2019), we use adaptive inputs and an adaptive softmax (Grave et al., 2017b) with tied weights (Press & Wolf, 2017) for the WIKITEXT-103 experiments. On other datasets we do not use adaptive inputs or an adaptive softmax.

**Evaluation**    LMs are trained to minimize the negative log-likelihood of the training corpus, and evaluated by perplexity (exponentiated negative log-likelihood) on held out data. Following Baevski & Auli (2019), 512 tokens are scored per test example, but up to 2560 tokens of extra prior context is provided for WIKITEXT-103 and up to 512 tokens of extra prior context is provided for the rest of the corpora.

$k$**NN-LM**    The keys used for $k$NN-LM are the 1024-dimensional representations fed to the feed-forward network in the final layer of the Transformer LM (after self-attention and layernorm; see Section 5 for further explanation). We perform a single forward pass over the training set with the trained model, in order to save the keys and values. During this forward pass, each target token is provided a minimum of 1536 tokens of prior context for WIKITEXT-103 and a minimum of 512

| Model | Perplexity (↓) | | # Trainable Params |
|---|---|---|---|
| | Dev | Test | |
| Baevski & Auli (2019) | 17.96 | 18.65 | 247M |
| +Transformer-XL (Dai et al., 2019) | - | 18.30 | 257M |
| +Phrase Induction (Luo et al., 2019) | - | 17.40 | 257M |
| Base LM (Baevski & Auli, 2019) | 17.96 | 18.65 | 247M |
| +$k$NN-LM | **16.06** | **16.12** | 247M |
| +Continuous Cache (Grave et al., 2017c) | 17.67 | 18.27 | 247M |
| +$k$NN-LM + Continuous Cache | **15.81** | **15.79** | 247M |

Table 1: Performance on WIKITEXT-103. The $k$NN-LM substantially outperforms existing work. Gains are additive with the related but orthogonal continuous cache, allowing us to improve the base model by almost 3 perplexity points with no additional training. We report the median of three random seeds.

| Model | Perplexity (↓) | | # Trainable Params |
|---|---|---|---|
| | Dev | Test | |
| Base LM (Baevski & Auli, 2019) | 14.75 | 11.89 | 247M |
| +$k$NN-LM | **14.20** | **10.89** | 247M |

Table 2: Performance on BOOKS, showing that $k$NN-LM works well in multiple domains.

tokens for the rest of the corpora. A FAISS index is then created using 1M randomly sampled keys to learn 4096 cluster centroids. For efficiency, keys are quantized to 64-bytes. During inference, we retrieve $k = 1024$ neighbors, and the index looks up 32 cluster centroids while searching for the nearest neighbors. For WIKITEXT-103 experiments, we compute squared $L^2$ distances with full precision keys, but for the other datasets we use the FAISS $L^2$ distances (not squared) between quantized keys directly, for faster evaluation. We tune the interpolation parameter $\lambda$ on the validation set.[1]

**Computational Cost**    Although the $k$NN-LM requires no training given an existing LM, it does add some other computational overheads. Storing the keys and values requires a single forward pass over the training set, which amounts to a fraction of the cost of training for one epoch on the same examples. Once the keys are saved, for WIKITEXT-103 building the cache with 103M entries takes roughly two hours on a single CPU. Finally, running on the validation set took approximately 25 minutes when retrieving 1024 keys. While the cost of building a large cache grows linearly in the number of entries, it is trivial to parallelize and requires no GPU-based training.

## 4    EXPERIMENTS

### 4.1    USING THE TRAINING DATA AS THE DATASTORE

We first experiment with creating a datastore from the same data used to train the LM. Table 1 shows that $k$NN-LM improves perplexity on WIKITEXT-103 from 18.65 (Baevski & Auli, 2019) to a new state-of-the-art of 16.12. We also provide reported perplexities from two other recent models that also build upon Baevski and Auli's, suggesting that further improvements may be possible by augmenting the $k$NN-LM with these techniques. We compare with models trained only on the standard training set, but recent work has shown performance can be improved by training on additional data, from either the test set (Krause et al., 2019) or large amounts of web text (Shoeybi et al., 2019).

We also experiment with a continuous cache model, a related but orthogonal technique from Grave et al. (2017c), in which the model saves and retrieves neighbors from earlier in the test document,

---

[1]Code is available at: `https://github.com/urvashik/knnlm`

| Training Data | Datastore | Perplexity (↓) | |
| --- | --- | --- | --- |
| | | Dev | Test |
| WIKI-3B | - | 16.11 | 15.17 |
| WIKI-100M | - | 20.99 | 19.59 |
| WIKI-100M | WIKI-3B | 14.61 | 13.73 |

Table 3: Experimental results on WIKI-3B. The model trained on 100M tokens is augmented with a datastore that contains about 3B training examples, outperforming the vanilla LM trained on the entire WIKI-3B training set.

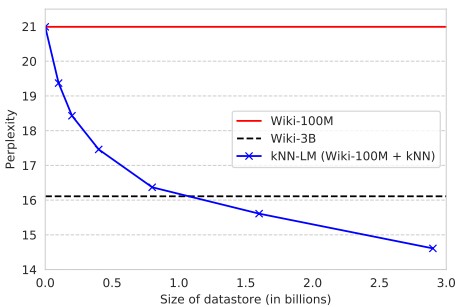

(a) Effect of datastore size on perplexities.

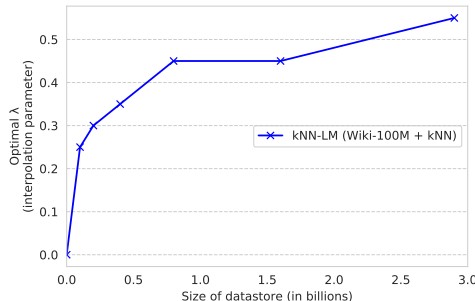

(b) Tuned values of $\lambda$ for different datastore sizes.

Figure 2: Varying the size of the datastore. (a) Increasing the datastore size monotonically improves performance, and has not saturated even at about 3B tokens. A $k$NN-LM trained on 100M tokens with a datastore of 1.6B tokens already outperforms the LM trained on all 3B tokens. (b) The optimal value of $\lambda$ increases with the size of the datastore.

rather than the training set. Gains from interpolating with the continuous cache are smaller than reported in the original setting that used LSTMs, perhaps because self-attentive language models can learn to perform such queries. Improvements from the continous cache are additive with the $k$NN-LM, pushing our state-of-the-art result to 15.79, a gain of 2.86 over the base model.

Finally, we repeat the experiment using text from a different domain, BOOKS, to control for the possibility that encyclopedic Wikipedia text is somehow uniquely good for caching. Table 2 shows an improvement in test set perplexity from 11.89 to 10.89, suggesting that this is not the case.

## 4.2 MORE DATA WITHOUT TRAINING

Section 4.1 has shown that retrieving neighbors from the training data can significantly improve language modeling performance. This raises the question: can retrieving nearest neighbors from data be a substitute for training on it? To test this, we train a LM on WIKI-100M and use it to build a datastore from WIKI-3B, a corpus 30 times larger than the training set. We then compare this $k$NN-LM to a vanilla LM trained on the entire WIKI-3B corpus.[2]

Table 3 shows that, as expected, the model trained on 3B tokens dramatically outperforms the model trained on 100M tokens, improving perplexity from 19.59 to 15.17. However, adding nearest neighbors retrieval over those 3B examples to the model trained on 100M tokens improves perplexity from 19.59 to 13.73; i.e. *retrieving nearest neighbors from the corpus outperforms training on it*. This result suggests that rather than training language models on ever larger datasets, we can use smaller datasets to learn representations and augment them with $k$NN-LM over a large corpus.

---

[2]The original LM (Baevski & Auli, 2019) was trained for 286K steps on a corpus of similar size to WIKI-100M. When scaling up to WIKI-3B, we tuned only the number of updates on the validation set and found that training for 572K steps (double) produces a slightly stronger baseline.

| Training Data | Datastore | Perplexity (↓) | |
|---|---|---|---|
| | | Dev | Test |
| WIKI-3B | - | 37.13 | 34.84 |
| BOOKS | - | 14.75 | 11.89 |
| WIKI-3B | BOOKS | 24.85 | 20.47 |

Table 4: Domain adaptation experiments, with results on BOOKS. Adding an in-domain datastore to a Wikipedia-trained model improves results by 23 points, approaching in-domain training.

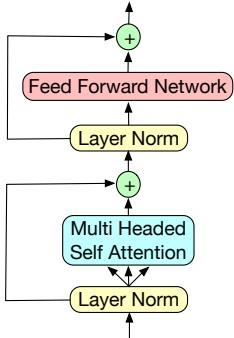

Figure 3: Transformer LM layer.

| Key Type | Dev ppl. (↓) |
|---|---|
| No datastore | 17.96 |
| Model output | 17.07 |
| Model output layer normalized | 17.01 |
| FFN input after layer norm | **16.06** |
| FFN input before layer norm | 17.06 |
| MHSA input after layer norm | 16.76 |
| MHSA input before layer norm | 17.14 |

Table 5: WIKITEXT-103 validation results using different states from the final layer of the LM as the representation function $f(\cdot)$ for keys and queries. We retrieve $k$=1024 neighbors and $\lambda$ is tuned for each.

To understand how the amount of data used for $k$NN retrieval affects performance, we use the WIKI-100M model to create datastores using different amounts of randomly sampled data from WIKI-3B. Figure 2a shows that using only 1.6B examples for the datastore already surpasses the performance of the model trained on all of WIKI-3B. In addition, performance does not saturate at 3B examples in the datastore, suggesting that growing the datastore more could lead to further gains. Figure 2b shows the model relies more on the $k$NN component as the size of the datastore increases.

## 4.3 DOMAIN ADAPTATION

We also experiment with domain adaptation by creating a datastore on the target domain training set. Table 4 shows that an in-domain LM on BOOKS has a relatively low perplexity (11.89), while a model trained on WIKI-3B performs poorly on the BOOKS domain (34.84 perplexity). Adding $k$NN search over BOOKS to the WIKI-3B model reduces perplexity by 14 points (to 20.47), demonstrating that $k$NN-LM allows a single model to be useful in multiple domains, by simply adding a datastore per domain.

## 5 TUNING NEAREST NEIGHBOR SEARCH

While the $k$NN-LM is conceptually straightforward, and requires no additional training, a number of hyperparameters are introduced for nearest neighbor search. We experiment with different choices here.

**Key Function** For similarity search, we extract a representation of context $c$ using an intermediate state of the LM $f(c)$. Transformers compute a number of different intermediate states, and we compare several choices depicted in Figure 3, with results shown in Table 5. While all the instantiations of $f$ we tried are helpful, we achieved the largest improvement by using the input to the final layer's feedforward network. We also observe that normalized representations (i.e. taken immediately after the layer norm) perform better. Repeating the experiment on the second-last transformer layer showed similar trends with slightly worse results (not shown), suggesting that the feedforward layer might be focusing more on the prediction problem, while the onus of representing the input falls more on the self-attention layer.

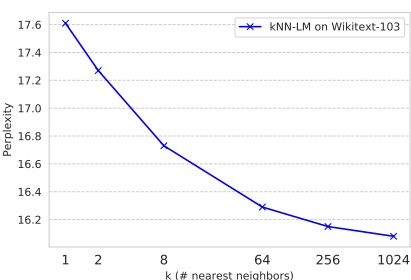
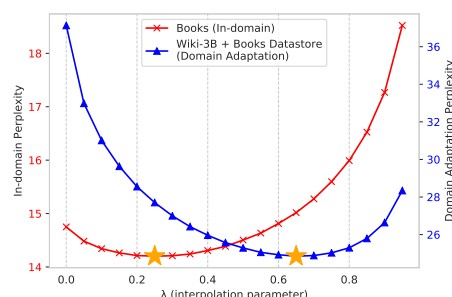

Figure 4: Effect of the number of nearest neighbors returned per word on WIKITEXT-103 (validation set). Returning more entries from the datastore monotonically improves performance.

Figure 5: Effect of interpolation parameter $\lambda$ on in-domain (left y-axis) and out-of-domain (right y-axis) validation set performances. More weight on $p_{kNN}$ improves domain adaptation.

**Number of Neighbors per Query**   Each query returns the top-$k$ neighbors. Figure 4 shows that performance monotonically improves as more neighbors are returned, and suggests that even larger improvements may be possible with a higher value of $k$. Nonetheless, even a small number of neighbors ($k = 8$) is enough to achieve a new state of the art.

**Interpolation Parameter**   We use a parameter $\lambda$ to interpolate between the base model distribution and the distribution from $k$NN search over the dataset. Figure 5 shows that $\lambda = 0.25$ is optimal on WIKITEXT-103. However, $\lambda = 0.65$ works best for domain adaptation results (Figure 5).

**Precision of Similarity Function**   In FAISS, the nearest neighbor search computes $L^2$ distances against quantized keys. We found results were improved from 16.5 perplexity on WIKITEXT-103 to 16.06 by computing squared $L^2$ distances with full precision keys for Equation 2.

## 6   ANALYSIS

**Qualitative Analysis**   To understand why $k$NN-LM improves performance, we manually examine cases in which $p_{kNN}$ was significantly better than $p_{LM}$. Table 6 shows one such example, along with several others in Appendix A. The example shows an interesting case where the model matches the trigram *impact on the* in several retrieved neighbors, but puts almost all weight on the most relevant neighbor, thus adding more value than an $n$-gram LM.

In general, we find that examples where $k$NN-LM is most helpful typically contain rare patterns. Examples include factual knowledge, names, and near-duplicate sentences from the training set. In these cases, assigning train and test instances similar representations (via $f(\cdot)$) appears to be an easier problem than implicitly memorizing the next word in model parameters.

**Simple vs Neural Representation**   We observe that many long-tail phenomena manifest as rare $n$-grams (e.g. names). Is it therefore possible to interpolate an $n$-gram model with a Transformer LM, as an alternative to our $k$NN approach? Figure 7 shows little improvement from using $n$-gram LMs – 0.2 perplexity points (similarly to Bakhtin et al. (2018)). This result highlights the need to use the learned representation function $f(\cdot)$ to measure similarity between more varied contexts.

**Implicit vs Explicit Memory**   If a neural representation function is crucial for $k$NN-LM, could implicitly memorizing the training dataset in the neural network parameters replace the explicit memory in the datastore? To test this, we train a Transformer LM with no dropout. Figure 8 shows that this model eventually reaches zero training loss, indicating that it can make perfect predictions for all examples in the training set; the model has memorized the dataset. Naturally, the memorizing LM overfits, i.e. the training loss drops to 0 while the best validation perplexity is much higher at 28.59. For comparison, the vanilla Transformer LM (with dropout) has a much higher training loss (shown in Figure 8), but also generalizes better with a validation perplexity of 17.96. This result shows that the Transformer has sufficient capacity to memorize the training set.

| Test Context | ($p_{\text{kNN}} = 0.998$, $p_{\text{LM}} = 0.124$) | Test Target |
|---|---|---|
| *it was organised by New Zealand international player Joseph Warbrick, promoted by civil servant Thomas Eyton, and managed by James Scott, a publican. The Natives were the first New Zealand team to perform a haka, and also the first to wear all black. They played 107 rugby matches during the tour, as well as a small number of Victorian Rules football and association football matches in Australia. Having made a significant impact on the...* | | development |

| Training Set Context | Training Set Target | Context Probability |
|---|---|---|
| *As the captain and instigator of the 1888-89 Natives – the first New Zealand team to tour the British Isles – Warbrick had a lasting impact on the...* | development | 0.998 |
| *promoted to a new first grade competition which started in 1900. Glebe immediately made a big impact on the...* | district | 0.00012 |
| *centuries, few were as large as other players managed. However, others contend that his impact on the...* | game | 0.000034 |
| *Nearly every game in the main series has either an anime or manga adaptation, or both. The series has had a significant impact on the...* | development | 0.00000092 |

Figure 6: Example where the $k$NN model has much higher confidence in the correct target than the LM. Although there are other training set examples with similar local $n$-gram matches, the nearest neighbour search is highly confident of specific and very relevant context.

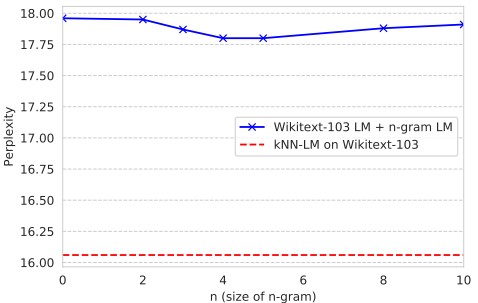

Figure 7: Interpolating the Transformer LM with $n$-gram LMs on WIKITEXT-103 (validation set). Using $k$NN-LM gives a much lower perplexity, suggesting that the representations are learning more than just matching local context.

Figure 8: Training curves for the Transformer LM with and without dropout. Turning off dropout allows the training loss to go to 0, indicating that the model has sufficient capacity to memorize the training data.

We consider whether the memorizing LM can be an effective substitute for nearest neighbor search. Interpolating the memorizing LM with the original LM improves validation perplexity by just 0.1 – compared to 1.9 from $k$NN-LM. This result suggests that although the Transformer is expressive enough to memorize all training examples, learning to do so does not result in context representations that generalize. In contrast, $k$NN-LM memorizes training data while improving generalization.

From these experiments, we conjecture that $k$NN-LM improves performance because (1) the Transformer LM is very good at learning a representation function for contexts with an implicit notion of similarity, and (2) while the Transformer has capacity to memorize all training examples, doing so causes its representation to generalize less effectively, but (3) the $k$NN-LM allows the model to memorize the training data while retaining an effective similarity function.

## 7 RELATED WORK

We discuss related uses of caches for language modeling in Section 2.

Similar $k$NN models to ours have been proposed for computer vision tasks (Papernot & McDaniel, 2018; Orhan, 2018; Zhao & Cho, 2018), primarily motivated by improving interpretability and robustness to adversarial attacks. We hypothesize that our method may be particularly effective for language modeling, because plentiful unlabeled data allows datastores of billions of tokens, and language modeling often requires world knowledge to be learnt from few examples.

Nearest neighbor models have been applied to a number of NLP problems in the past, such as part of speech tagging (Daelemans et al., 1996) and morphological analysis (Bosch et al., 2007), but the use of learned representations makes the similarity function much more effective in the case of neural models. More recently, Kaiser et al. (2017) have used a similarly differentiable memory that is learned and updated during training, and is applied to one-shot learning tasks.

Several models have also improved language generation by using training examples directly at test time. Guu et al. (2018) propose a model that samples training sentences at random and edits them with a sequence-to-sequence model, but does not use a retrieval mechanism such as $k$NN. Gu et al. (2018) introduce a translation model that attends over retrieved training set examples. Weston et al. (2018) improve a dialogue response generation model by refining similar instances from the training set. $k$NN-LM differs from these approaches by working at the level of individual tokens instead of whole training sentences, as well as not incorporating the retrieval mechanism into the training pipeline.

A general trend in machine learning, and in language modeling in particular, is that adding more data consistently improves performance (Devlin et al., 2019; Radford et al., 2019; Yang et al., 2019; Liu et al., 2019; Zellers et al., 2019; Shoeybi et al., 2019). Our work offers an alternative method for scaling language models, in which relatively small models learn context representations, and a nearest neighbour search acts as a highly expressive classifier.

## 8 CONCLUSION AND FUTURE WORK

We have introduced $k$NN-LMs, which can significantly outperform standard language models by directly querying training examples at test time. The approach can be applied to any neural language model. The success of this method suggests that learning similarity functions between contexts may be an easier problem than predicting the next word from some given context. Future work should explore explicitly training similarity functions, and reducing the size of the datastore.

### ACKNOWLEDGMENTS

The authors thank the anonymous reviewers as well as Sida Wang, Kartikay Khandelwal, Kevin Clark and members of the FAIR Seattle team for helpful discussions and comments.

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

# A APPENDIX

This section provides several examples where $p_{kNN}$ places higher probability mass on the true target, compared to $p_{LM}$.

| Test Context    ($p_{kNN} = 0.995$, $p_{LM} = 0.025$) | Test Target | |
|---|---|---|
| *For Australians and New Zealanders the Gallipoli campaign came to symbolise an important milestone in the emergence of both nations as independent actors on the world stage and the development of a sense of national identity. Today, the date of the initial landings, 25 April, is known as Anzac Day in Australia and New Zealand and every year thousands of people gather at memorials in both nations, as well as Turkey, to...* | honour | |
| **Training Set Context** | **Training Set Target** | **Context Probability** |
| *Despite this, for Australians and New Zealanders the Gallipoli campaign has come to symbolise an important milestone in the emergence of both nations as independent actors on the world stage and the development of a sense of national identity. Today, the date of the initial landings, 25 April, is a public holiday known as Anzac Day in Australia and New Zealand and every year thousands of people gather at memorials in both nations, and indeed in Turkey, to ...* | honour | 0.995 |
| *On the anniversary date of his death, every year since 1997, thousands of people gather at his home in Memphis to...* | celebrate | 0.0086 |
| *Twenty-five years after Marseille's death, fighter pilot veterans of World War II gathered to...* | honour | 0.0000041 |

Table 6: Another example where the $k$NN model places much higher probability mass on the correct target, compared to the LM. The nearest neighbors search has retrieved a training set context that is extremely similar to the test context, while very rare and in the long-tail of patterns.

| Test Context    ($p_{kNN} = 0.959$, $p_{LM} = 0.503$) | Test Target | |
|---|---|---|
| *U2 do what they're best at, slipping into epic rock mode, playing music made for the arena". In two other local newspaper reviews, critics praised the song's inclusion in a sequence of greatest hits. For the PopMart Tour of 1997–...* | 1998 | |
| **Training Set Context** | **Training Set Target** | **Context Probability** |
| *Following their original intent, "Sunday Bloody Sunday" was not played during any of the forty-seven shows on the Lovetown Tour in 1989. The song reappeared for a brief period during the Zoo TV Tour, and late during the second half of PopMart Tour (1997–...* | 1998 | 0.936 |
| *They are 6 times Champions and they won the Challenge Cup in 1938, and have experienced two previous stretches in the Super League, 1997–...* | 2002 | 0.0071 |
| *About \$40 million (\$61.4 million in 2018 dollars) was spent on the property acquisition. After weather-related construction delays due to the El Nino season of the winter of 1997–...* | 1998 | 0.0015 |
| *This made it the highest-rated season of The X-Files to air as well as the highest rated Fox program for the 1997–...* | 98 | 0.00000048 |

Table 7: In this example, the desired date pattern appears in many examples. Yet, the nearest neighbors search is able to identify the only training set context which is relevant to the test context and assigns it the highest probability mass.

| Test Context    ($p_{\mathrm{kNN}} = 0.624$, $p_{\mathrm{LM}} = 0.167$) | Test Target | |
|---|---|---|
| *Lord Strathcona awarded Gauthier a scholarship in 1906 that allowed her to return to Europe and continue her vocal studies. She returned there and continued both to study and give performances. Her first operatic performance came in 1909 in Pavia, Italy as Micaela in Bizet's...* | Carmen | |

| **Training Set Context** | **Training Set Target** | **Context Probability** |
|---|---|---|
| *Despite poor relations with the orchestra, Mahler brought five new operas to the theatre, including Bizet's...* | Carmen | 0.356 |
| *The fourth movement of An die Jugend (1909), for instance, uses two of Niccolo Paganini's Caprices for solo violin (numbers 11 and 15), while the 1920 piece Piano Sonatina No. 6 (Fantasia da camera super Carmen) is based on themes from Georges Bizet's...* | opera | 0.0937 |
| *It also hosted the Ballet of her Majesty's Theatre in the mid-19th century, before returning to hosting the London premieres of such operas as Bizet's...* | Carmen | 0.0686 |

Table 8: In this case, the model is able to memorize the fact that *Georges Bizet* wrote *Carmen*.

| Test Context    ($p_{\mathrm{kNN}} = 0.031$, $p_{\mathrm{LM}} = 0.007$) | Test Target | |
|---|---|---|
| *Mycena maculata bears some resemblance to M. <unk>, but is only associated with decaying hardwood logs and stumps, and is found in eastern North America, and sometimes on oak on the West Coast. In age, it...* | develops | |

| **Training Set Context** | **Training Set Target** | **Context Probability** |
|---|---|---|
| *Morchella tridentina (=Morchella frustrata) is also rufescent and very similar to M. rufobrunnea. It is found in mountainous forests and maquis and forms a marked sinus at the attachment of the cap with the stem, which is pure white. At maturity, it...* | develops | 0.031 |
| *The winter bonnet (M. tintinnabulum) is a northern European species that is much smaller (cap diameter up to 2.6 cm (1.0 in) across) and has a brown cap, and has ragged hairs at the base. It...* | generally | 0.029 |
| *The "bleeding" will distinguish Mycena atkinsoniana from most other Mycena species commonly encountered. The common and widely distributed M. sanguinolenta is another "bleeder", but it is smaller than M. atkinsonia, with a cap diameter ranging from 3 to 15 mm (0.1 to 0.6 in). Additionally, it...* | has | 0.028 |
| *Mycena flavoalba bears resemblance to some members of the genus Hemimycena, such as H. lactea and H. <unk>. It...* | can | 0.018 |

Table 9: This is an example where the $p_{\mathrm{kNN}}$ distribution is relatively flat, as several words are plausible continuations. However, the nearest neighbors search assigns the highest probability to the correct target and a corresponding context that is particularly relevant. In contrast, the LM probability on the correct target is lower.

