# OpenReview forum: "Generalization through Memorization: Nearest Neighbor Language Models"
_ICLR.cc/2020/Conference — Accept (Poster)_

### Official Review · AnonReviewer3 · 2019-10-18
**Official Blind Review #1318**

**Rating:** 3

**Review:**

This work utilizes the kNN method on dense vectors to augment the LMs. The method is simple and straightforward, meanwhile, the performance seems great if only in terms of PPL.
Three of my most concerns:
1)	It seems that this approach heavily relies on the similarity of context distribution between the training and test set. Intuitively, higher performance will be achieved with more similar examples between training and test set. This question should be discussed more in this work. This similarity cannot always satisfied in practice, I thus quite doubt the proposed method can work for general case.
2)	The evaluation is only done for PPL, I notice the LM was trained in a corpus scale as pre-trained BERT, though none of real downstream tasks were evaluated like BERT. Expect to see some MRC or NLI results with the proposed LM.
3)	Furthermore, though FAISS is very fast, it is hard to get great results with only a small datastore which makes the retrieving slow. So it seems not suitable for tasks such as generations but maybe open-domain QA can be the scene for this method. It would be great if there are some experiments on such tasks, and also combining with models such as BERT could be much better and convincing.

Questions
What about other distance functions such as cosine distance? The author only said L2 is better but there is no analysis on it.


**Experience Assessment:**

I have published one or two papers in this area.

**Review Assessment: Checking Correctness Of Derivations And Theory:**

I assessed the sensibility of the derivations and theory.

**Review Assessment: Checking Correctness Of Experiments:**

I carefully checked the experiments.

**Review Assessment: Thoroughness In Paper Reading:**

I read the paper thoroughly.

---

> ### Public Comment · ~Felix_Wu1 · 2019-11-06
> **There might be some misunderstanding... this is not a representation learning paper**
>
> Hi AnonReviewer3,
>
> There might be some misunderstanding regarding your second point.
> This is not a representation learning paper. The authors propose to apply kNN on top of a pre-trained LM which requires no additional training. Evaluations on downstream tasks (like MRC or NLI) are not applicable in their case.
> However, this is totally understandable. I also had this wrong impression when I read the paper for the first time.
>
> Also, I believe it's common in the LM community to report just PPL (or BPC for character-level LM).
> For example, Baevski & Auli (ICLR 2019) and Dai et. al. (ACL 2019).
> Admittedly, I personly would be interested in seeing kNN-LM being applied to other generation tasks where they can use other metrics such as ROUGE-L for summarization. I imagine this may be a followup paper.

---

> > ### Author Response · Authors · 2019-11-08
> > **Thanks, Felix!**
> >
> > Thanks for your clarification, Felix!! We absolutely agree with your points and will be sure to add an explicit note to the paper highlighting the fact that this work is not related to learning better contextual representations. Thanks to you and Reviewer3, we were able to catch the ambiguity and can fix it early!
> >
> > We also agree the approach could naturally be applied to summarization and translation, and think this would be an exciting direction for future work.

---

> > ### Comment · AnonReviewer3 · 2019-11-09
> > **It is LM training on large text, both are problematic on model size and PPL**
> >
> > Thanks for your task purpose clarification, Alex, though I have been always understanding that this work is an extra LM enhancement component.
> > I would like to make me more clear here on my stand. My focus is still on the metric and the evaluation.
> > (1)[This part is something new after I read the other comments, including those from the authors]
> > I fully understand PPL is a well-accepted task-independent metric for LMs. Lower PPL on test has shown the proposed kNN improvement somewhat works. However, such performance improvement may not be from the idea of kNN, or the memory mechanism, but actually from much more parameters hidden in the datastore (which is supposed very large no matter preprocessing is done on it or not). The kNN is good, and the memory mechanism is also good. However, I really not appreciate either of them brings a too large storage component. More parameters for NN or other machine learning methods usually show better performance, I am not impressed by this.
> > Note that Table 1 claims there is no parameter increasing as the proposed component is applied, which is an unfair comparison. Even though the datastore cannot be seen additional parameters, the proposed kNN module has greatly increased the model (or system) size for running.
> > Besides, from my regard, all LMs, either traditional n-gram LM, or the latest pre-trained ones, are actually to do a job that compress their huge training corpus, so that we can have a concise enough LM to be used. I do not think it is practical to use the proposed kNN enhancement in real NLP tasks. Pre-trained LMs with 1G (BERT) or 3G (XLNet) model size have been regarded too large for real computation, then how about the proposed model?
> > (2)[This part is still about insufficiency of PPL only evaluation.]
> > I have trained ELMo from zero by myself, and also quite deeply fine-tune BERT, which made me distrust PPL too much. As this paper has reported, PPL can be always decreasing on training curve (only if the training lasts). My experience on ELMo training also shows the same trend. Though all these belong to training performance evaluation, which cannot reflex PPL on test set or the generalized ability. However, LM training is a specific machine learning task, it is supposed to learn general linguistic knowledge from as large corpus as possible. When SOTA pre-trained LMs have trained on dozens of billions of text, I do not think the difference between training PPL or test PPL really matters, as to some extent, billons of text has covered all reasonably collected text generated by human kind. Keep it in mind that training curve of PPL is always down only if the training lasts.
> > All latest pre-trained LMs have to show their effectiveness on various downstream NLP tasks, which I guess it from the same reason that PPL cannot be a sufficient evaluation metric.
> > Overall, LMs are a basic support tool for all later NLP tasks. Thus it is actually not about the pre-trained, or non-pre-trained, as evaluation on downstream tasks has been a routine treatment in LM work, I find no reason why the proposed method cannot report their evaluation results in the same way.

---

> > > ### Author Response · Authors · 2019-11-12
> > > **Response to Reviewer3 comments**
> > >
> > > Hello Reviewer3,
> > >
> > > Thanks for your comments.
> > >
> > > Convergence and Evaluation:
> > > Regarding the concerns about convergence and evaluation, we note that the autoregressive LM used in this work has been trained to convergence for every experiment, i.e. the dev loss is seen going up at the end of training. For such LMs (vastly different from bi-directional masked LMs like BERT), it is standard practice to evaluate performance using perplexity (Shannon, 1951; Brown et al., 1992; Goodman, 2001; Bengio et al., 2003; Mikolov et al., 2010; Baevski and Auli, 2019, Dai et al., 2019 etc.), as also noted in Felix’s comment. Wikitext-103 is one of the standard benchmarks used for this task and all prior work has evaluated using perplexity as well.
> > >
> > > Large training sets:
> > > Regarding large training sets, we note that two of our experiments used models trained to convergence on 3-billion tokens of Wikipedia, (1) Table 3 where we show training on so much data is perhaps unnecessary and a kNN-LM can be far more effective instead, and (2) Table 4 where we show even a model trained on this much data is not very effective at domain adaptation, while a kNN-LM makes this model useful in multiple domains and hence better at generalization.
> > >
> > > Cost:
> > > While the kNN component does require storage it is not GPU based, which makes this storage very cheap. This is possible because the kNN component does not add any trainable parameters and requires no further parameter updates on the GPU, unlike larger models. Querying this module is also fast using the FAISS library, which allows a reasonable decoding speed of 60 tokens per second using the Wikitext-103 datastore containing 100-million entries.
> > >
> > > Train/test distributions:
> > > For cases where the test set distribution is vastly different from the LM training set distribution, our domain adaptation results have shown how kNN-LM helps improve generalization (shown in Table 4). For cases where the test set distribution is vastly different from both the LM training set and the datastore, we expect model generalization to be no worse than that of the base LM already.
> > >
> > > We do agree that applying this model to tasks such as translation and summarization, as well as further research into reducing the size of the kNN datastore all make for exciting next steps.
> > >
> > > Thanks!

---

### Official Review · AnonReviewer2 · 2019-10-23
**Official Blind Review #2**

**Rating:** 6

**Review:**


Summary:
The authors extend a pretrained LM by interpolating its next word distribution with a KNN model. The authors show retrieving nearest neighbor from corpus achieve quite large perplexity decrease in several language modeling benchmarks.

Decision:
Overall, the idea seems simple but is quite effective. Even with some discussions on the related work with cache based LM and the work that use training examples explicitly, I feel it is a simple extension/usage of previous approaches. Hence I am borderline with my decision.

Supporting argument:
1. The proposed idea uses KNN to look up training examples for interpolating the prediction. As discussed by the authors, this approach is effective in factual knowledge, names, and near-duplicate sentences.
2. There are several experiments and ablation study in showing the effectiveness of the approach.
3. The related work that uses training examples explicitly is quite similar to the proposed approach, though the authors claim that one is at the level of individual tokens and the other is the whole training sentences.

Additional feedback:
1. In reference, ‘Bert’ -> ‘BERT’
2. Missing reference: Yogatama et al., Memory Architectures in Recurrent Neural Network Language Models, 2018, https://arxiv.org/abs/1410.3916, https://arxiv.org/abs/1803.02400

**Experience Assessment:**

I have read many papers in this area.

**Review Assessment: Checking Correctness Of Derivations And Theory:**

N/A

**Review Assessment: Checking Correctness Of Experiments:**

I assessed the sensibility of the experiments.

**Review Assessment: Thoroughness In Paper Reading:**

I made a quick assessment of this paper.

---

> ### Author Response · Authors · 2019-11-12
> **Response to Reviewer2**
>
> Hello Reviewer2,
>
> Thanks for your comments.
>
> As per your suggestion, we’ll add more details comparing our method against related work! The difference between kNN-LM and prior work is certainly larger than just operating at the token vs. the sentence level: prior work uses training examples very differently. Guu et al. (2018) sample a training example at random and edit it into a new sentence. Gu et al. (2018) look up training examples using edit distance against the test string that needs to be translated. Both Gu et al. (2018) and Weston et al. (2018) train their models with the retriever and use the embeddings retrieved as inputs to the model. In contrast, our kNN module requires no training and uses retrieved examples as the model’s prediction directly.
>
> This model highlights the effectiveness of the similarity function that is learned by the LM. In fact, our work shows that instead of using large models trained on large datasets, we may be able to use smaller models that learn effective similarity functions to generalize to larger datasets as well as to other domains, without any additional training necessary. This sets us on an exciting path of thinking about using kNN to make our models more effective without necessarily scaling them up!
>
> Thanks for your notes on the memory networks literature. We’re working on adding a contrast there as well as a note on work from Walter Daelemans on pre-neural memory based language processing (2005)!
>
> Thanks!

---

### Official Review · AnonReviewer1 · 2019-10-26
**Official Blind Review #1**

**Rating:** 6

**Review:**

[Overview]

In this paper, the authors proposed a simple but effective way to augmentation the language model through memorization. Specifically, after obtaining a language model on a dataset, the model further uses the dataset to build a lookup table and then a k-nearest neighbor is used to searching the closest tokens for a token during inference. Based on this, the output distribution of a target token during the inference time would be modified accordingly. Through a comprehensive experiments and ablation studies, the authors showed that the proposed strategy can improve the performance of language models significantly for both the in-domain and out-domain testing scenarios. This is very insightful considering recently a lot of language models are focusing on increasing the size of model and training data.

[Pros]:

Overall I think the paper is well-written and presents clearly. Detailed points below:

1. the authors proposed a simple but effective method for increasing the generalization ability of language model through a memorization strategy. Specifically, the authors proposed to build a lookup table which memorizes the representation and output token pairs which are then used for the inference of language model. Different from conventional way, the proposed strategy does not introduce any more parameters in the model and also does not need any more training or fine-tuning on the target dataset.

2. The authors showed that the proposed strategy can improve the performance of language generation model (i.e., transformer) without any extra training or data, as shown in Table 1. Also, using the continuous caches  with KNN-LM further improve the performance.

3. Besides the main results shown in Table 1 and Table 2, the authors also showed using kNN-LM can probably outperforms the model which is directly trained on it. Also, it also supports domain adaptation from one language domain to another domain.

4. Finally, the authors presented a number of ablation studies to investigate how the performance is affected by the method of building datastore, including the size of nearest neighbor, the interpolation parameter, etc. These results are also insightful and meaningful for the readers to understand the method.

[Cons]:

I think this paper is a solid paper. So I would have some suggestions below:

1. The first concern about the method is the efficiency. At page 3, the authors mentioned that the proposed strategy will bring more time cost. It would be good if the authors can perform more systematical analysis on the time cost of building the datastore and inference for the proposed model.

2. Second, the authors should not only evaluate the proposed method based on transformers. It would be good to test on various language models to verify the generalization ability across different models, including the old-fashioned one like RNN and CNN.

3. Also, the authors should try to extend the proposed model to other language tasks, such as translation.

[Summary]

In this paper, the authors introduced a simple but effective method to augment the pertained language model through memorizations. Though this is not absolutely new and relatively simple , the authors successfully demonstrate that it can be applied to improve the generation of language model much. The. thorough ablation studies help to understand the property of the proposed strategy. I think this paper overall is insightful and thoughtful. It would be good to see the authors add more analysis on the computational complexity and also evaluate on more type of language models.


**Experience Assessment:**

I have read many papers in this area.

**Review Assessment: Checking Correctness Of Derivations And Theory:**

I assessed the sensibility of the derivations and theory.

**Review Assessment: Checking Correctness Of Experiments:**

I assessed the sensibility of the experiments.

**Review Assessment: Thoroughness In Paper Reading:**

I read the paper at least twice and used my best judgement in assessing the paper.

---

> ### Author Response · Authors · 2019-11-12
> **Response to Reviewer1**
>
> Hello Reviewer1,
>
> Thanks for your comments. We’re glad you enjoyed the paper!
>
> Efficiency:
> Building the datastore: A single epoch of training over the Wikitext-103 data takes ~5 hours on a single GPU. In comparison, a single forward pass over the same dataset to save keys/values took ~4 hours. Then, creating the datastore using FAISS took two hours on a single CPU. Hence, building the datastore is is comparable to a single epoch of training. In addition, the saving of keys/values as well as creating the datastore are trivial to parallelize.
>
> Inference: We measured the decoding speed of kNN-LM and found that it can sample roughly 60 tokens per second on one GPU, which is easily fast enough for most applications (albeit slower than the vanilla LM, which can sample roughly 500 tokens per second). Improving the efficiency is not a focus of this work, but it is likely that it could be significantly improved - for example, by downsampling frequent words from the datastore.
>
> Other architectures:
> We are in the process of evaluating the model on a CNN-based LM as well! Thanks for the suggestion!
>
> Future work:
> We also agree that applying kNN-LM to translation would be an exciting next step which we hope to pursue in followup work!
>
> Thanks!

---

> > ### Author Response · Authors · 2019-11-15
> > **Follow-up**
> >
> > Hello Reveiwer1, we just wanted to follow up on our previous comment about generating results for a CNN-based model. Our experiments are currently running, but unfortunately we won’t have the results before the end of the discussion period.

---

### Author Response · Authors · 2019-10-11
**Updated Results**

Hello Reviewers,

Since submission, we realized that the Books-1B dataset contained duplicate books in train/test, which our model is particularly effective at exploiting. To mitigate this effect, we re-ran these experiments on a de-duplicated version of Books-1B, and present updated tables below. Results on other datasets are unchanged because those were already de-duplicated, and all conclusions still hold.

Table 2 Books-1B:
Baevski & Auli (2019) 	                Dev = 14.75 		Test = 11.89
kNN-LM 			                Dev = 14.20 		Test = 10.89

Table 4 Domain adaptation for Books-1B:
Wiki-3B				                Dev = 37.13		Test = 34.84
Books-1B				        Dev = 14.75		Test = 11.89
Wiki-3B + Books-1B datastore	Dev = 24.85		Test = 20.47

Given the above results, our conclusions still follow. Table 2 shows that kNN-LM helps in domains other than Wikipedia. Table 4 shows that while an in-domain LM trained on Books-1B has relatively low perplexity (11.89), an LM trained on Wiki-3B and evaluated on Books-1B performs considerably worse (34.84). Adding a datastore containing Books-1B training examples to the Wiki-3B model reduces perplexity by 14 points (down to 20.47) demonstrating that kNN-LM allows a single model to be useful in multiple domains without additional training.

Thanks!

---

### Public Comment · ~Jack_William_Rae1 · 2019-11-04
**Question about model ensembles**

I really enjoyed reading this paper and think the execution is very impressive. It certainly wasn't clear to me that a KNN would benefit the test perplexity for a dataset like WikiText-103 and so it feels like there's an important story to tell here, alongside the great results.

One train of thought is that the two models composed jointly are a small ensemble model; does this perform much better than if you had two separately trained transformer language models ensembled, tuned on the validation set? That is, how much of the benefit can we be sure is from the differences of the two models, versus the general benefit of ensembling? To take that idea further, one could imagine ensembling a regular transformer language model with an overfitted transformer that behaves like a KNN (because it has memorized the training set). By an overfit model, I mean... Train a 24 layer model without dropout on WikiText-103 until it is close to 1 training perplexity. Would this setup behave just as well (or better?). If usual ensembling doesn't help, but ensembling between an LM + overfit LM does help, then that would give further evidence to the benefit of ensembling diverse models.

---

> ### Author Response · Authors · 2019-11-04
> **Model ensembles**
>
> Hi Jack,
>
> Thanks for your comments!
>
> You raise an interesting question! We actually included a version of this experiment in the submission: by turning off dropout in our base (16 layer) model, it reached a training perplexity of 1 (see Figure 8). This result shows that the 16 layer model does have the capacity to memorize the entire training set in its parameters.
>
> However, ensembling the overfit model with the original LM only improved the validation perplexity by 0.1 (see discussion in Section 6). This suggests that while the Transformer certainly has the capacity to memorize the training set, doing so does not result in context representations that generalize well enough to mimic kNN-LM’s explicit nearest neighbors mechanism.
>
> Thanks!

---

> > ### Public Comment · ~Jack_William_Rae1 · 2019-11-05
> > **Re.**
> >
> > Thanks for clarifying my ignorance in missing the 0.1 improvement result in Section 6! I did see you had trained an overfit model in Figure 8 but I didn't see that you had mixed it with your original LM to get a measly 0.1 improvement --- this is one of the most convincing results in the paper IMO. It's interesting that two different memorization schemes can have very different generalization properties. I suppose the entropy of the output probabilities of the overfit model is a fair bit lower than the KNN - they both memorize but the KNN retains a bit more uncertainty. I still think it could be interesting to calculate the perplexity from ensembling two early-stopped transformer models also to further explore the model-ensemble benefits vs model diversity benefits; but of course we operate in a domain of limited time.

---

> > > ### Public Comment · ~Qianying_Lin2 · 2022-03-23
> > > **Interesting point**
> > >
> > > I think this paper is very interesting.  I agree with Jack on the possibility to compare against an ensemble model between the overfit model and the original model, as this intuitively works better than ensemble with the KNN-based method. As far as I understand, the overfit Transformer with 0 training loss obtains the highest accuracy possible on rare samples for the training set. The KNN-based method is a compromise since we cannot re-score all samples when training each sample. Therefore, with an appropriate interpolation hyper-parameter, it should theoretically work better? Also I conjecture there could possibly be one possible optimization point : maybe there could be a counter counting the frequency of patterns, and uses a smaller $\lambda$ on the original LM model for the less frequent patterns?

---

### Public Comment · ~Aurko_Roy1 · 2019-11-06
**Datastore construction**

Really cool paper - I enjoyed the idea of combining language modeling with k-nearest neighbor retrieval! I had a question about the datastore construction - the same token might have different contexts appearing in the corpus. In that case do you have repeated entries for the same token with two different keys? Or do you do some sort of averaging? Apologies if I missed this in the paper.

---

> ### Author Response · Authors · 2019-11-08
> **Datastore**
>
> Hi Aurko,
>
> Thanks for your comments!
>
> There is no averaging! We save every context-target pair. So the same word appears many times in the datastore each with different keys corresponding to the different contexts it appeared in. To compute the final kNN probability of a word, we aggregate over all occurrences of that word retrieved in the k-nearest neighbors set.
>
> Thanks!

---

### Decision · Program_Chairs · 2019-12-19

**Decision:**

Accept (Poster)

**Comment:**

This paper proposes an idea of using a pre-trained language model on a potentially smaller set of text, and interpolating it with a k-nearest neighbor model over a large datastore. The authors provide extensive evaluation and insightful results. Two reviewers vote for accepting the paper, and one reviewer is negative. After considering the points made by reviewers, the AC decided that the paper carries value for the community and should be accepted.